# A picorna-like virus suppresses the N-end rule pathway to inhibit apoptosis

Zhaowei Wang[1,2†], Xiaoling Xia[1,2,3†], Xueli Yang[1], Xueyi Zhang[1,2], Yongxiang Liu[1,2], Di Wu[1,2], Yuan Fang[1,2], Yujie Liu[1,2], Jiuyue Xu[2], Yang Qiu[2], Xi Zhou[1,2]*

[1]State Key Laboratory of Virology, College of Life Sciences, Wuhan University, Wuhan, China; [2]State Key Laboratory of Virology, Wuhan Institute of Virology, Chinese Academy of Sciences, Wuhan, China; [3]Guangzhou Key Laboratory of Insect Development Regulation and Application Research, Institute of Insect Science and Technology & School of Life Sciences, South China Normal University, Guangzhou, China

**Abstract** The N-end rule pathway is an evolutionarily conserved proteolytic system that degrades proteins containing N-terminal degradation signals called N-degrons, and has emerged as a key regulator of various processes. Viruses manipulate diverse host pathways to facilitate viral replication and evade antiviral defenses. However, it remains unclear if viral infection has any impact on the N-end rule pathway. Here, using a picorna-like virus as a model, we found that viral infection promoted the accumulation of caspase-cleaved *Drosophila* inhibitor of apoptosis 1 (DIAP1) by inducing the degradation of N-terminal amidohydrolase 1 (NTAN1), a key N-end rule component that identifies N-degron to initiate the process. The virus-induced NTAN1 degradation is independent of polyubiquitylation but dependent on proteasome. Furthermore, the virus-induced N-end rule pathway suppression inhibits apoptosis and benefits viral replication. Thus, our findings demonstrate that a virus can suppress the N-end rule pathway, and uncover a new mechanism for virus to evade apoptosis.
DOI: https://doi.org/10.7554/eLife.30590.001

*For correspondence:
zhouxi@wh.iov.cn

†These authors contributed equally to this work

Competing interests: The authors declare that no competing interests exist.

## Introduction

Apoptosis is a highly conserved biological process throughout evolution and is important for normal tissue development and removal of obsolete, abnormal or potentially harmful cells. The apoptotic pathway shows sensitivity to various stimuli and can lead to cysteinyl aspartate protease (caspase) dependent proteolytic digestion and further cell death (*Benedict et al., 2002*; *Kumar, 2007*). Various viruses, including vertebrate and invertebrate viruses, can induce apoptosis in infected cells or organisms (*Everett and McFadden, 1999*; *Lamiable et al., 2016*; *Lannan et al., 2007*; *Liu et al., 2013*; *Nainu et al., 2015*; *Settles and Friesen, 2008*). Apoptosis is generally considered as an efficient antiviral defense mechanism by clearing virus-infected cells, while many viruses employ different strategies to evade apoptosis at various levels (*Benedict et al., 2002*; *Everett and McFadden, 1999*; *Kim et al., 2014a*, *2013*).

The fruit fly *Drosophila melanogaster* has made a great contribution to study the regulation of apoptosis. Similar to other organisms, the caspase proteases are the central executioners of apoptosis in *Drosophila*. The fly caspase-9 homolog Dronc is the only known initiator caspase that is activated following a variety of apoptotic stimuli and can be activated by auto-cleavage (*Muro et al., 2004*). Activated Dronc can further cleave and activate various effector caspases such as DrICE and DCP-1, leading to apoptotic induction. The initiator caspase Dronc and the effector caspases DrICE

and DCP-1 are negatively regulated by DIAP1 (*Hawkins et al., 1999*, *2000*; *Li et al., 2011*; *Meier et al., 2000*; *Wilson et al., 2002*).

DIAP1 shares several properties in structure and function with mammalian X-linked inhibitor of apoptosis (XIAP) and can block cell death in response to multiple stimuli (*Hay et al., 1995*). As a central cell death regulator, DIAP1 is regulated by several distinct ways. For instance, *Drosophila* Reaper, Hid and Grim (also referred to RHG proteins) can inhibit the apoptosis suppression activity of DIAP1 or induce the degradation of DIAP1 (*Huh et al., 2007*; *Wang et al., 1999*; *Yoo et al., 2002*). Besides, DIAP1 can be auto-ubiquitylated via its C-terminal RING ubiquitin ligase domain (*Wilson et al., 2002*) or be ubiquitylated by other E3 ubiquitin ligases such as DIAP2 (*Herman-Bachinsky et al., 2007*), followed by proteasome-dependent degradation. It has also been reported that DIAP1 can be degraded by the N-end rule pathway. In this process, DIAP1 is cleaved at Asp20 by caspase to expose an N-terminal Asn residue. The exposed N-terminal Asn can be recognized and converted into Asp by NTAN1, and further catalyzed by Arginine-tRNA-protein transferase (ATE1) (*Ditzel et al., 2003*). Such Arg-conjugated proteins can be recognized and ubiquitylated by the N-end rule specific E3 ubiquitin ligase, UBR1, and then subject to fast degradation (*Ditzel et al., 2003*).

The N-end rule pathway is a proteasome dependent proteolytic system that recognizes and degrades proteins containing N-degrons (*Gibbs et al., 2014a*; *Tasaki et al., 2012*; *Varshavsky, 2011*; *Tasaki and Kwon, 2007*). This pathway has been found to be evolutionarily conserved from prokaryotic to eukaryotic organisms, including bacteria (*Tobias et al., 1991*), yeast (*Bachmair et al., 1986*), plant (*Graciet et al., 2009*; *Yoshida et al., 2002*), invertebrate (*Ditzel et al., 2003*), and vertebrate (*Davydov and Varshavsky, 2000*; *Lee et al., 2005*; *Park et al., 2015*). The N-end rule pathway relates the half-lives of proteins with the nature of their N-termini (*Gibbs et al., 2014a*; *Tasaki et al., 2012*; *Varshavsky, 2011*; *Tasaki and Kwon, 2007*). A functional N-degron can either be an unmodified destabilizing N-terminal residue or an N-terminally modified (deamidated, oxidized, and/or arginylated) pre-N-degron (*Varshavsky, 2011*; *Tasaki and Kwon, 2007*). In the case of DIAP1, caspase cleaves DIAP1 to expose an N-terminal Asn residue (*Ditzel et al., 2003*). This Asn residue is a classical pre-N-degron for N-terminal deamidation by NTAN1, followed by arginylation by ATE1. It has been reported that the N-end rule pathway participates in a large number of important cellular processes, such as G protein signaling (*Davydov and Varshavsky, 2000*; *Lee et al., 2005*; *Park et al., 2015*), chromosome stability (*Rao et al., 2001*), apoptosis (*Ditzel et al., 2003*), oxygen and nitric oxide sensing (*Gibbs et al., 2014b*), degradation of neurodegeneration-associated protein fragments (*Brower et al., 2013*) and etc. Moreover, the N-end rule pathway has been reported to interact with some viral proteins. For instance, Sindbis virus nsP4 and HIV-1 integrase are N-end rule substrates (*de Groot et al., 1991*; *Mulder and Muesing, 2000*), and human papillomavirus E7 binds to UBR4, the E3 ligase in the N-end rule pathway (*White et al., 2012*). However, it remains unclear if viral infection has any impact on this pathway.

Here, we report that the infection by a picorna-like virus can induce apoptosis in infected *Drosophila* cells, and the apoptotic pathway plays an antiviral role in *Drosophila*. Intriguingly, we found that the viral infection promoted the accumulation of caspase-cleaved, smaller form of DIAP1, which is potent for apoptosis inhibition, by inhibiting the N-terminal Asn deamidation of the cleaved DIAP1. Moreover, we uncovered that the viral infection could induce the degradation of NTAN1, which catalyzes the N-terminal Asn deamidation of the cleaved, smaller DIAP1. And the virus-induced NTAN1 degradation is independent of polyubiquitylation but dependent on proteasome. Furthermore, our study revealed that the virus-induced N-end rule pathway suppression could efficiently block apoptosis and facilitates viral replication. In summary, our findings demonstrate for the first time that a virus can suppress the N-end rule pathway, and uncover a new mechanism for virus to evade apoptosis.

## Results

### Viral infection induces apoptosis in *Drosophila*

Previous studies showed that various viruses, including *Autographa californica* nucleopolyhedrovirus (AcMNPV), Flock House Virus (FHV), and *Drosophila* C virus (DCV), can induce apoptosis in *Drosophila* cells or adult flies (*Lamiable et al., 2016*; *Lannan et al., 2007*; *Liu et al., 2013*; *Nainu et al.,*

*2015*; *Settles and Friesen, 2008*). Among these viruses, DCV, which is a picorna-like virus assigned to the family *Dicistroviridae* of the order *Picornavirales*, is a natural pathogen of *Drosophila* and a classic model virus (*Johnson and Christian, 1998*). To confirm whether DCV infection can also induce apoptosis in our system, we performed a flow cytometry assay using Annexin V-allophycocyanin (APC)/propidium iodide (PI) double staining in cultured *Drosophila* S2 cells. Annexin V staining can detect the surface exposure of phosphatidylserine, a hallmark of apoptosis, while PI staining can identify dead cells. Consistent with previous study (*Lamiable et al., 2016*), DCV-infected cells showed increased Annexin V and PI staining as infection progressed when comparing with mock infected cells (*Figure 1A and B*). Moreover, we used terminal deoxynucleotide transferase-mediated dUTP nick-end labeling (TUNEL) staining to detect apoptotic cells. In this assay, DCV-infected cells also showed an increase in apoptotic cell death comparing with mock infected cells (*Figure 1C*). In addition, previous study has reported that the transcriptions of RHG genes were up-regulated by the AcMNPV or FHV infection in adult flies (*Liu et al., 2013*). Our data showed that DCV infection induced RHG gene transcription in *Drosophila* S2 cells (*Figure 1D*). The level of *reaper* mRNA was significantly induced at 6 hr post infection (h.p.i) of DCV, while a significant induction of *hid* or *grim* mRNA can be detected at 12 h.p.i (*Figure 1D*). Altogether, DCV infection is able to induce the transcription of RHG genes and apoptosis in cultured *Drosophila* cells.

## Inhibition of apoptosis enhances viral replication in cells and adult flies

After determining that DCV infection induces apoptosis, we further examined whether apoptosis has any antiviral role. To this end, we ectopically expressed DIAP1 in cultured S2 cells to inhibit apoptosis. Our results showed that the ectopic expression of DIAP1 effectively inhibited apoptosis (*Figure 2A*) and caused about two-fold increase of DCV genomic RNA (*Figure 2B*). Moreover, when we knocked down both of the effector caspases DrICE and DCP1, the virus-induced apoptosis was also dramatically inhibited (*Figure 2D*), resulting in a significant increase of DCV genomic RNA in infected cells (*Figure 2E*). In addition, the inhibition of apoptosis by either DIAP1 overexpression or effector caspases knockdown similarly increased DCV genomic RNA levels in cultured fluids (*Figure 2C and F*), excluding the possibility that the increase of DCV genomic RNA levels in cells is caused by promoting virus entry or inhibiting virus release.

To assess whether apoptosis contributes to inhibit viral replication in adult flies, we performed a DCV oral infection assay using p53 loss-of-function fly allele 5A-1–4 ($p53^{-/-}$). This fly allele has a reduced level of stress-induced apoptosis, but is otherwise viable and has no obvious phenotype (*Liu et al., 2013*; *Rong et al., 2002*). 7 days after the DCV oral infection, almost all $p53^{-/-}$ flies were dead, while about 40% control flies survived in the viral challenge (*Figure 2G*). These data indicate that the loss of p53 function made adult flies more susceptible to viral infection. We further tested the DCV genomic RNA level at 3 days post DCV oral infection, and observed approximately 5-fold increase of DCV genomic RNA in $p53^{-/-}$ flies, when comparing with control flies (*Figure 2H*). Taken together, our data show that apoptosis plays an antiviral role in cultured *Drosophila* S2 cells and adult flies.

## Viral infection promotes the accumulation of cleaved DIAP1 in cells

As one of the most important cell death regulators, DIAP1 has been reported to be depleted during the course of FHV infection (*Settles and Friesen, 2008*). To study whether DCV infection has any effect on DIAP1, we determined the levels of endogenous DIAP1 using western blot in DCV-infected *Drosophila* S2 cells. DIAP1 has been gradually depleted during the course of DCV infection (*Figure 3A*). To examine whether DCV infection promotes DIAP1 degradation, cycloheximide (CHX) degradation assays have been conducted. Because CHX treatment can efficiently block viral protein synthesis and viral replication, we infected cells using DCV immediately after or 8 hr before adding CHX. Although viral infection immediately after CHX addition did not accelerate DIAP1 depletion (*Figure 3—figure supplement 1A*, lanes 1 and 2 vs. 3 and 4), viral infection before CHX addition did promote DIAP1 degradation (*Figure 3—figure supplement 1A*, lanes 5 and 6, 1B, 1C, and 1D). These results show that DCV infection promotes the degradation of DIAP1, and this process relies on viral protein synthesis and/or viral replication, but not the input viral components, as blocking viral protein synthesis eliminated this effect.

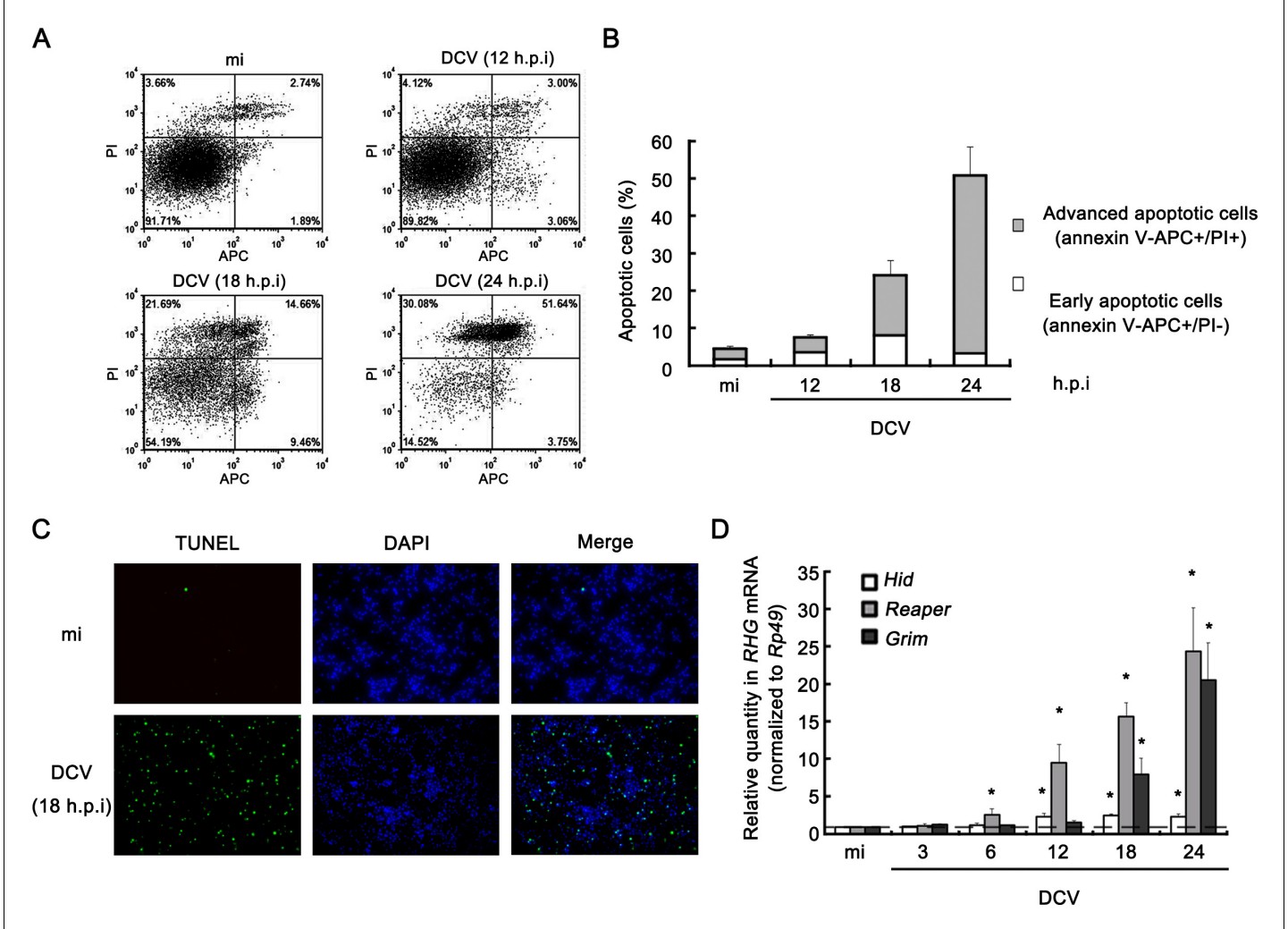

**Figure 1.** Viral infection induces apoptosis in *Drosophila* S2 cells. (**A**) Cultured S2 cells were mock infected for 24 hr or infected with DCV (MOI = 5) for indicated time. Annexin-V-APC/PI double staining and flow cytometry assay was performed to quantify viable (Annexin-V-APC-/PI-), early apoptotic (Annexin-V-APC+/PI-) and late apoptotic cells (Annexin-V-APC+/PI+). (**B**) The percentage of early apoptotic cells and late apoptotic cells after mock infected for 24 hr or infected with DCV (MOI = 5) for indicated time (*n* = 3; error bars, s.d.). (**C**) S2 cells were mock infected or infected with DCV (MOI = 5) for 18 hr and analyzed by a TUNEL assay. Detection of DNA using DAPI staining was performed in the same experiment. TUNEL+ signals are green and DAPI+ signals are blue. (**D**) Cultured S2 cells were mock infected for 24 hr or infected with DCV (MOI = 5) for indicated time. After that, total RNA extracts were prepared for qRT-PCR assay of hid, reaper or grim mRNA (normalized to Rp49; *n* = 3; error bars, s.d.). mi, mock infection.
DOI: https://doi.org/10.7554/eLife.30590.002

The following source data is available for figure 1:

**Source data 1.** Quantification data for *Figure 1*.
DOI: https://doi.org/10.7554/eLife.30590.003

Intriguingly, a slightly smaller, faster-migrating form of endogenous DIAP1 can be detected (*Figure 3A*), leading us to ask how this smaller form of DIAP1 was generated. As illustrated in *Figure 3B*, a smaller form of DIAP1 can be either produced by caspase cleavage at Asp20 or by internal initiation at an in-frame second ATG (*Ditzel et al., 2003*; *Vandergaast et al., 2015*; *Vandergaast et al., 2011*). To distinguish between these two mechanisms, we first used the pancaspase inhibitor z-VAD-FMK to block the caspase activity. Our result showed that the presence of z-VAD-FMK could effectively block the production of the smaller DIAP1 (*Figure 3C*). Additionally, we also knocked down DrICE or DCP-1 by RNA interference (RNAi). Consistent with the results in

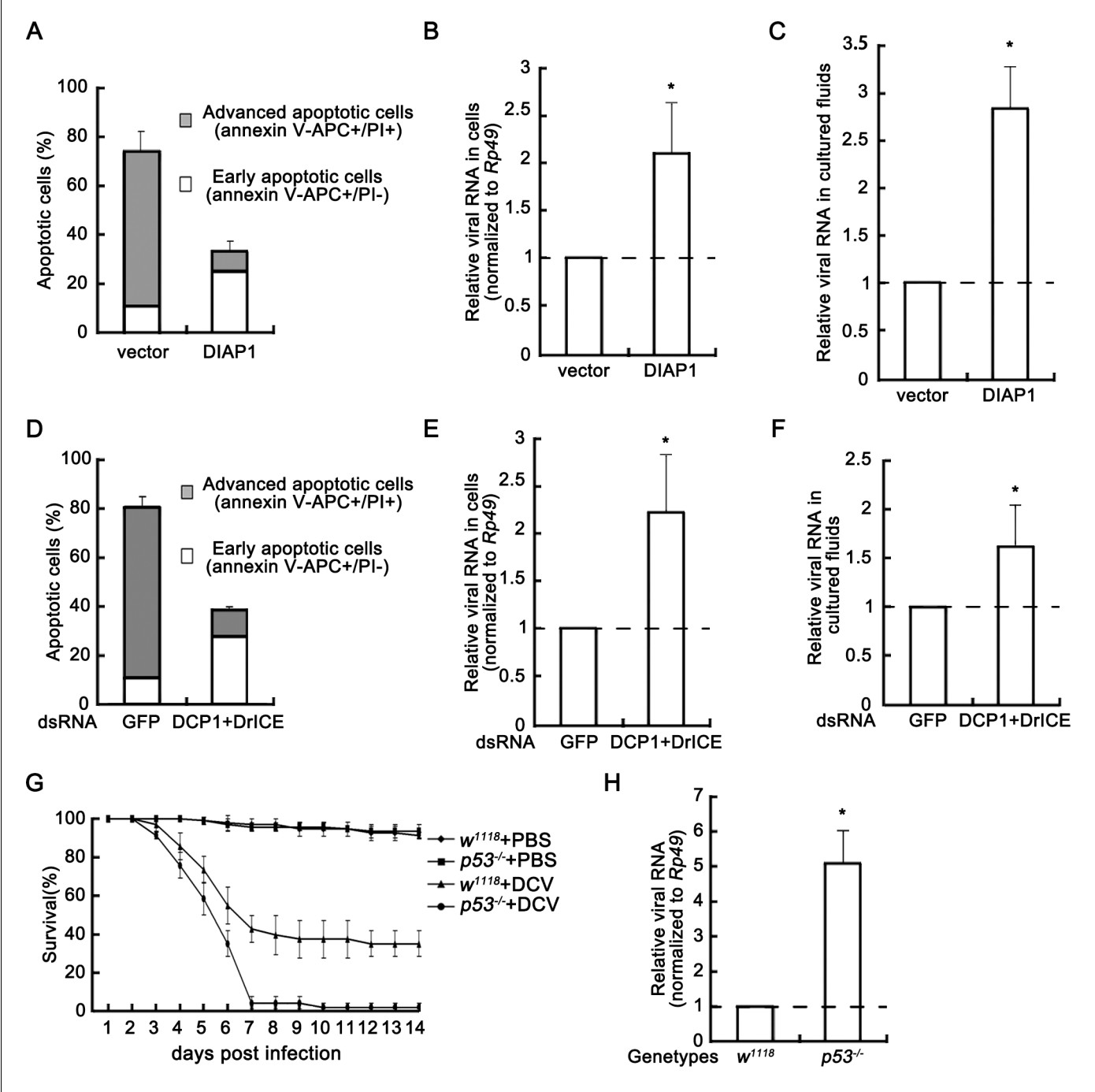

**Figure 2.** Apoptosis plays an antiviral role. (A) Cultured S2 cells were transfected with empty vector or the plasmid expressing DIAP1 as indicated, and then infected with DCV (MOI = 5) for 24 hr. The percentages of early apoptotic and late apoptotic cells were measured by Annexin-V-APC/PI double staining and flow cytometry assay (n = 3; error bars, s.d.). (B–C) Cultured S2 cells were transfected and infected as described in (A). After that, total RNAs in cells (B) and in 5% of cultured fluids (C) were extracted, followed by qRT-PCR assay of viral genomic RNA (n = 3; *, p<0.05 by two-tailed Student's t test; error bars, s.d.). For (B), viral genomic RNAs were normalized to Rp49. (D) Cultured S2 cells were transfected with dsRNAs against indicated genes and then infected with DCV (MOI = 5) for 24 hr. The percentages of early apoptotic and late apoptotic cells were measured by Annexin-V-APC/PI double staining and flow cytometry assay (n = 3; error bars, s.d.). (E–F) Cultured S2 cells were transfected and infected as described in (D). After that, total RNAs in cells (E) and in 5% of cultured fluids (F) were extracted, followed by qRT-PCR assay of viral genomic RNA (n = 3; *p<0.05 by two-tailed Student's t test; error bars, s.d.). For (E), viral genomic RNAs were normalized to Rp49. (G) Survival of adult flies with indicated genotypes after DCV ($10^{11.5}$ TCID50/ml) oral infection or mock infection (n = 3; each group contains 15 female flies and 15 male flies; error bars, s.d.). (H) Total

*Figure 2 continued on next page*

*Figure 2 continued*

RNA extracts from adult flies with indicated genotypes after DCV ($10^{11.5}$ TCID50/ml) oral infection for 3 days were prepared for qRT-PCR assay of viral genomic RNA (normalized to Rp49, $n = 3$; *p<0.05 by two-tailed Student's $t$ test; error bars, s.d.).

DOI: https://doi.org/10.7554/eLife.30590.004

The following source data is available for figure 2:

**Source data 1.** Quantification data for *Figure 2*.

DOI: https://doi.org/10.7554/eLife.30590.005

*Figure 3C*, the knockdown of either effector caspase DrICE or DCP-1 mostly blocked the appearance of the smaller DIAP1 (*Figure 3D*).

Next, we ectopically expressed DIAP1 with N-terminal myc tag and C-terminal HA tag (myc-DIAP1-HA) in cultured S2 cells. As expected, a smaller form of exogenously expressed DIAP1 was readily detected using anti-HA but not anti-myc antibody (*Figure 3E*), showing that this smaller form of DIAP1 lost its N-terminal. This smaller form of exogenously expressed DIAP1 accumulated during the course of viral infection, but was almost completely degraded at 24 h.p.i. (*Figure 3E*). Of note, qRT-PCR assays have been used to confirm that the samples had the same levels of transfection (*Figure 3—figure supplement 2*). We further used z-VAD-FMK to block the caspase activity, and then detected the exogenously expressed DIAP1 using anti-HA antibody. Consistent with our previous data in *Figure 3C*, z-VAD-FMK blocked the generation of the smaller form of exogenously expressed DIAP1 (*Figure 3F*). These data indicate that the production of the smaller form of DIAP1 was mediated by caspase cleavage.

We then exogenously expressed the D20A mutant of DIAP1 (DIAP1$^{D20A}$), which cannot be cleaved by caspase (*Ditzel et al., 2003*). Our data showed that the D20A mutation eliminated the appearance of the smaller form of DIAP1 (*Figure 3G*). On the other hand, the other mutation, M38A, which blocks the internal initiation at the in-frame second ATG, failed to prevent the production of the smaller form of DIAP1, similarly with wild-type (WT) DIAP1 (*Figure 3H*).

It would be interesting to ask whether the cleaved, smaller form of DIAP1 is active in blocking apoptosis. To this end, we ectopically expressed a DIAP1 mutant DIAP1$^{\Delta N20}$, which loses its N-terminal 20 amino acid and mimics the cleaved form of DIAP1, in cells in the presence or absence of viral infection. Our data showed that the smaller form of DIAP1, DIAP1$^{\Delta N20}$, was also able to inhibit virus-induced caspase activity as effective as DIAP1$^{WT}$ (*Figure 3I*), indicating that this cleaved form of DIAP1 is still active.

In conclusion, our data show that viral infection caused the accumulation of a caspase-cleaved, smaller form of DIAP1, which is potent in apoptosis blockage, in cultured *Drosophila* cells.

## Virus-induced accumulation of cleaved DIAP1 is mediated by the N-end rule pathway

The accumulation of the caspase-cleaved, smaller form of DIAP1 during viral infection could be due to the enhancement in either caspase-mediated cleavage or protein stability. To distinguish between these two possibilities, we first determined the caspase activities during the course of viral infection. Interestingly, we observed that the caspase activity was enhanced after 12 h.p.i. (*Figure 4A*), while the apparent accumulation of the smaller DIAP1 was readily detectable at 6 h.p.i. (*Figure 3A*). Of note, the experiments in *Figures 3A* and *4A* were conducted using the same set of samples, excluding the possible variations of different samples. Thus, at least at early stage of viral infection, the accumulation of smaller DIAP1 is not due to enhanced caspase activity.

Next, we ought to examine whether the smaller DIAP1 accumulation is due to enhanced protein stability. As illustrated in *Figure 4B*, the caspase-cleaved, smaller form of DIAP1 can be degraded by different strategies as reviewed by Tasaki et al. (*Tasaki et al., 2012*), of which the N-end rule pathway is the only one specifically degrade the smaller DIAP1. Consistent with pervious study (*Ditzel et al., 2003*), knockdown of N-end rule pathway key component NTAN1 or ATE1 by RNAi resulted in the accumulation of caspase-cleaved, smaller DIAP1 (*Figure 4C*), confirming that the N-end rule pathway participates in the degradation of the caspase-cleaved DIAP1.

Moreover, we ectopically expressed either WT or N21A mutant DIAP1 in cells, as the N21A mutation makes the cleaved, smaller DIAP1 protein to lose its N-end Asn and become resistant to the

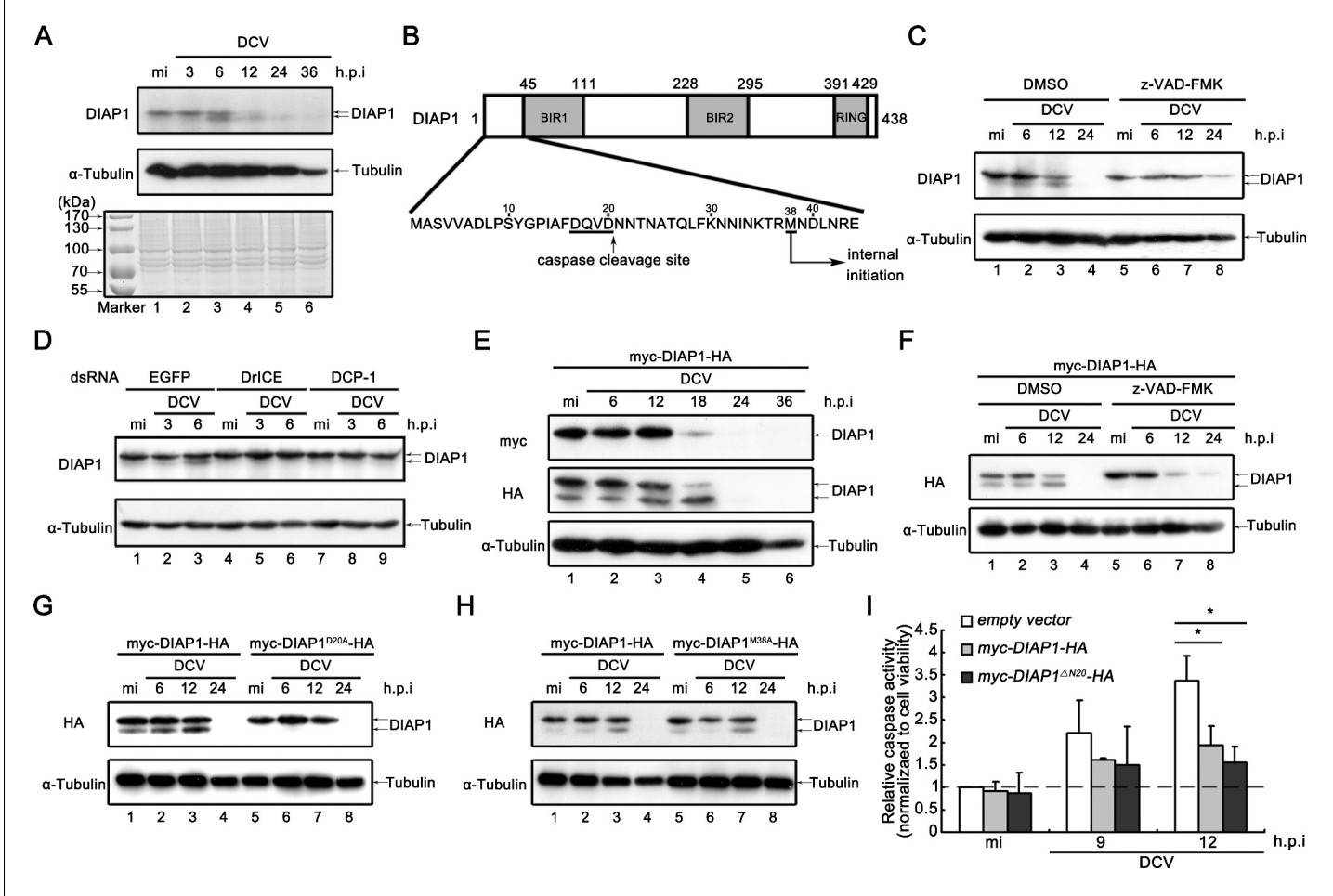

**Figure 3.** Viral infection promotes the accumulation of cleaved DIAP1 in *Drosophila* S2 cells. (**A**) Cultured S2 cells were mock infected for 36 hr or infected with DCV (MOI = 5) for indicated time. Cell lysates were subjected to SDS-PAGE, followed by western blots using the indicated antibodies or Coomassie Blue staining. (**B**) Schematic diagram of two distinct mechanisms to produce a smaller form of DIAP1. (**C**) Cultured S2 cells were treated with DMSO or z-VAD-FMK as indicated, and then mock infected for 24 hr or infected with DCV (MOI = 5) for indicated time. (**D**) Cultured S2 cells were transfected with dsRNAs against the indicated genes, and then mock infected for 6 hr or infected with DCV (MOI = 5) for indicated time. (**E**) Cultured S2 cells were transfected with plasmid expressing myc-DIAP1-HA, and then mock infected for 36 hr or infected with DCV (MOI = 5) for indicated time. (**F**) Cultured S2 cells were transfected with plasmid expressing myc-DIAP1-HA, and then treated with DMSO or z-VAD-FMK as indicated. After that cells were mock infected for 24 hr or infected with DCV (MOI = 5) for indicated time. (**G–H**) Cultured S2 cells were transfected with plasmid expressing myc-DIAP1-HA, myc-DIAP1$^{D20A}$-HA (**G**) or myc-DIAP1$^{M38A}$-HA (**H**) as indicated, and then mock infected for 24 hr or infected with DCV (MOI = 5) for indicated time. (**C–H**) Cell lysates were subjected to western blots using the indicated antibodies. (**I**) Cultured S2 cells were transfected with empty vector or plasmid expressing myc-DIAP1-HA or myc-DIAP1$^{\triangle N20}$-HA as indicated, and then mock infected or infected with DCV (MOI = 5) for indicated time. The relative caspase activity was measured, and normalized to cell viability (*n* = 3; *p<0.05 by two-tailed Student's *t* test; error bars, s.d.).

DOI: https://doi.org/10.7554/eLife.30590.006

The following source data and figure supplements are available for figure 3:

**Source data 1.** Quanitification data for *Figure 3* and *Figure 3—figure supplement 1.*
DOI: https://doi.org/10.7554/eLife.30590.009
**Figure supplement 1.** DCV infection promotes the degradation of DIAP1.
DOI: https://doi.org/10.7554/eLife.30590.007
**Figure supplement 2.** The DIAP1 overexpressed samples had the similar levels of transfection.
DOI: https://doi.org/10.7554/eLife.30590.008

N-end rule pathway. Our data show that although viral infection could induce the accumulation of smaller DIAP1 in cells expressing DIAP1$^{WT}$ from 0 (mock) to 15 h.p.i. (*Figure 4D*), the cleaved, smaller form of DIAP1 became resistant to degradation and insensitive to viral infection in

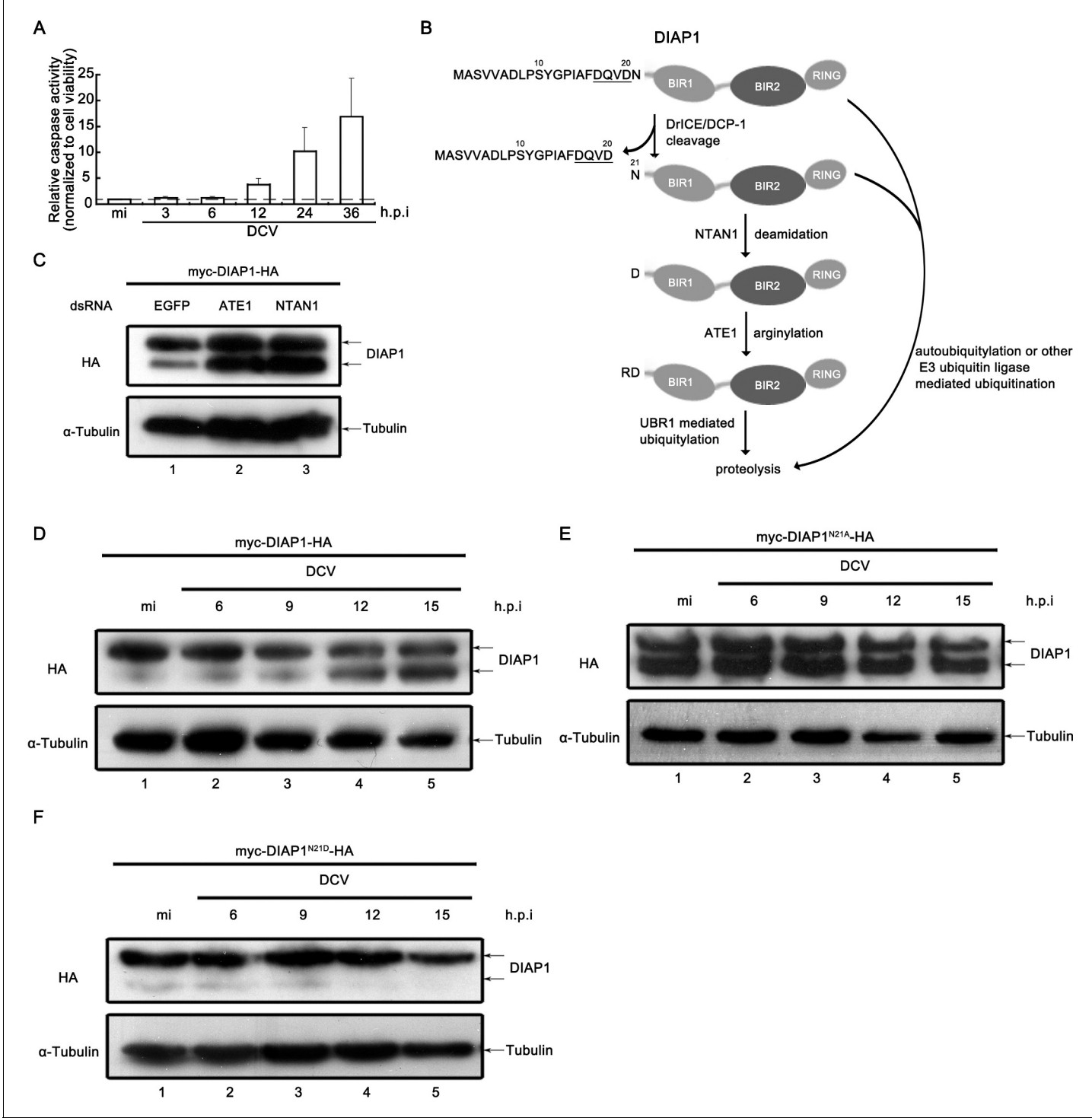

**Figure 4.** Viral infection inhibited the N-terminal Asn deamidation of cleaved DIAP1. (**A**) Cultured S2 cells were infected as described in *Figure 3A*. The relative caspase activity was measured, and normalized to cell viability ($n = 3$; *$p<0.05$ by two-tailed Student's $t$ test; error bars, s.d.). (**B**) Schematic diagram of the mechanisms of DIAP1 degradation. (**C**) Cultured S2 cells were transfected with the plasmid expressing myc-DIAP1-HA, and the dsRNAs against the indicated genes. (**D–F**) Cultured S2 cells were transfected with plasmid expressing myc-DIAP1-HA (**D**) or its mutants (**E and F**) as indicated, and then mock infected for 18 hr or infected with DCV (MOI = 5) for indicated time. (**C–F**) Cell lysates were subjected to western blots using the indicated antibodies.

DOI: https://doi.org/10.7554/eLife.30590.010

DIAP1$^{N21A}$-expressing cells (*Figure 4E*), showing that the effect of viral infection on the smaller DIAP1 accumulation is dependent on the N-end rule pathway.

Because the N-end rule pathway involves multiple steps, including deamidation by NTAN1, arginylation by ATE1, and proteolysis. We aim to investigate which step is affected by viral infection. To this end, we made the N21D mutation of DIAP1, which skips the N-terminal Asn deamidation step. Interestingly, viral infection did not increase the accumulation of the cleaved, smaller form of DIAP1$^{N21D}$ (*Figure 4F*), indicating that the inhibition of the deamidation step of the N-end rule pathway is required for the virus-induced accumulation of cleaved DIAP1.

## Viral infection promotes the depletion of NTAN1 in the early stage of infection

In the N-end rule pathway, the N-terminal Asn deamidation is catalyzed by NTAN1, while the arginylation of the deamidated protein is mediated by ATE1 (*Ditzel et al., 2003*). Consistent with our previous observation that the cleaved DIAP1 accumulation is dependent on the inhibition of NTAN1-mediated deamidation step (*Figure 4F*), our data show that viral infection induced the gradual decrease of the protein level of NTAN1 but not ATE1 (*Figure 5A*). Interestingly, the mRNA levels of NTAN1 and ATE1 are both up-regulated during the same time course of viral infection (*Figure 5B*). In addition, we examined the effect of viral infection to exogenously expressed NTAN1 in cultured cells, and found that the exogenously expressed NTAN1 was also down-regulated during the course of viral infection (*Figure 5C*). On the other hand, the level of exogenously expressed EGFP was not affected by viral infection (*Figure 5D*), confirming that the protein expression using the same expression vector was not affected by viral infection. Together, these data indicate that viral infection induced the decrease of NTAN1 protein level in a post-transcriptional manner.

To further assess whether the decrease of NTAN1 protein level during the course of viral infection is due to protein degradation, CHX degradation assays have been conducted. Similar with that of DIAP1, while viral infection immediately after CHX addition did not accelerate NTAN1 depletion (*Figure 5E*, lanes 1–2 vs. 3–4), viral infection before CHX addition significantly promoted NTAN1 degradation rate when compared with that in non-infected cells (*Figure 5E*, lanes 1–2 vs. 5–6, *Figure 5F,G and H*).

Interestingly, viral infection promoted the accumulation of NTAN1 after 12 h.p.i. (*Figure 5—figure supplement 1*), which might be a combined effect of both the degradation of NTAN1 protein and up-regulation of NTAN1 mRNA in the later stage of viral infection.

In conclusion, our data showed that virus induced the degradation of NTAN1 protein in the early stage of infection, which could lead to the accumulation of caspase cleaved, smaller form of DIAP1.

## Viral infection promotes the degradation of NTAN1 via the proteasome pathway

As we have found that viral infection promoted the degradation of NTAN1, we ought to investigate which protein degradation pathway(s) are involved in this process. Because the proteasome pathway is one of the major protein degradation pathways, we treated *Drosophila* S2 cells with proteasome inhibitor MG-132 or lactacystin. The results show that, during viral infection, the NTAN1 protein levels could be restored by either MG-132 or lactacystin treatment (*Figure 6A and B*), suggesting that the proteasome pathway is involved in virus-induced degradation of NTAN1.

Because the proteasome degradation pathway is usually dependent on polyubiquitylation, we next asked whether viral infection induces the polyubiquitylation of NTAN1. NTAN1 contains four lysine residues (i.e. K40, K63, K134 and K186). Among them, K186 is conserved in *Diptera* and vertebrate, K40 and K63 are conserved in *Diptera* but not vertebrate, while K134 is not conserved in *Diptera* (*Figure 6—figure supplement 1*). To investigate whether these residues are involved in the virus-induced NTAN1 degradation, we replaced all of the four lysine residues with alanine (NTAN1$^{4KA}$). However, when NTAN1$^{4KA}$ was exogenously expressed in *Drosophila* S2 cells, viral infection was still able to induce the decrease of NTAN1$^{4KA}$ protein level (*Figure 6C*). Furthermore, we conducted CHX degradation assay, and observed that in the absence of viral infection, NTAN1$^{4KA}$ is significantly more stable than NTAN1$^{WT}$, while viral infection similarly promoted the degradation of both NTAN1$^{4KA}$ and NTAN1$^{WT}$ (*Figure 6D–H*). These results show that the virus-induced NTAN1 degradation is independent of ubiquitylation.

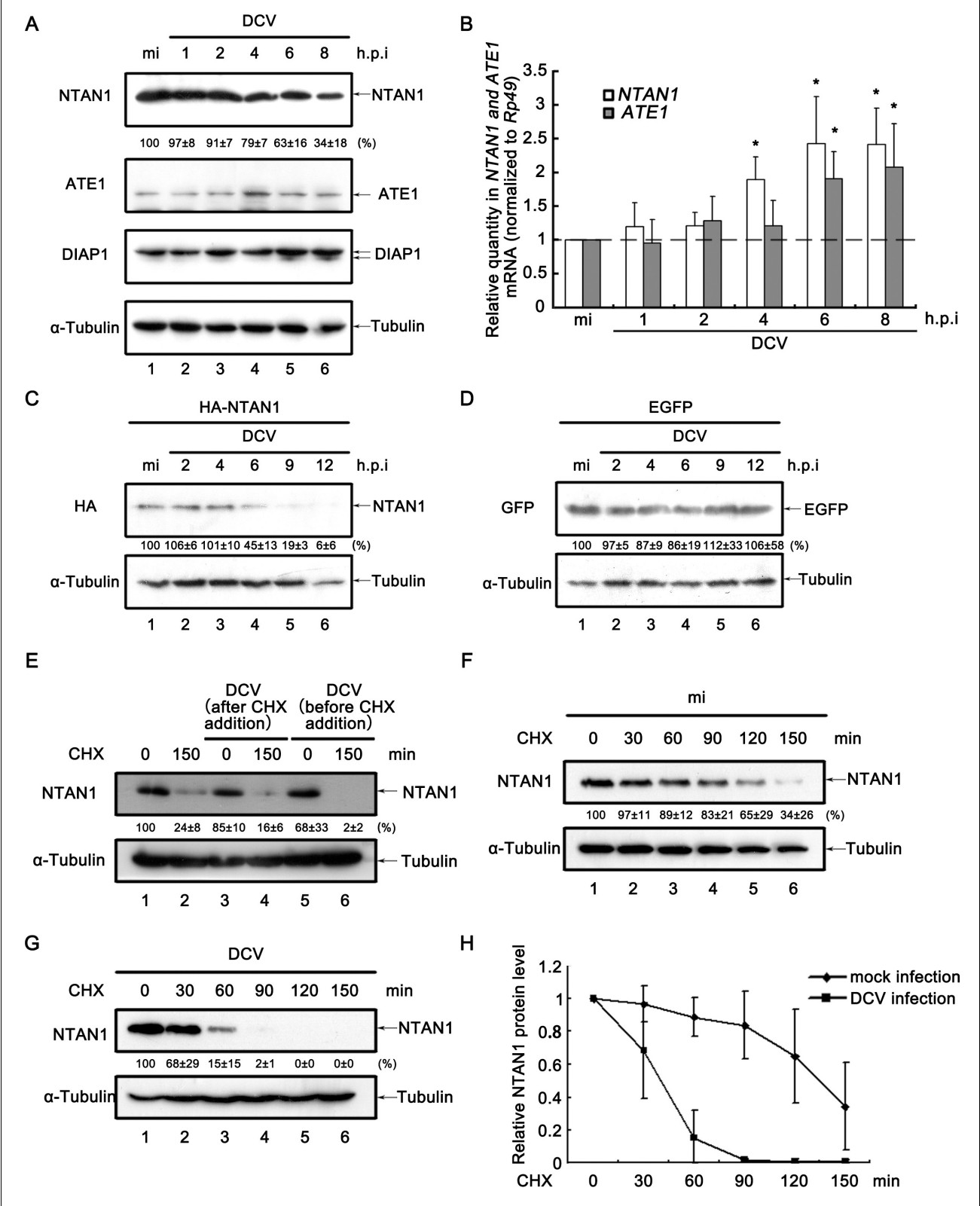

**Figure 5.** Viral infection promotes the degradation of NTAN1. (**A**) Cultured S2 cells were mock infected for 8 hr or infected with DCV (MOI = 5) for indicated time. Cell lysates were subjected to western blots using the indicated antibodies. (**B**) Cultured S2 cells were infected as described in (**A**). Total RNA extracts were prepared for qRT-PCR assay of indicated mRNA (normalized to Rp49; *n* = 3; error bars, s.d.). (**C–D**) Cultured S2 cells were transfected with plasmid expressing HA-NTAN1 (**C**) or EGFP (**D**) as indicated, and then mock infected for 12 hr or infected with DCV (MOI = 5) for

*Figure 5 continued on next page*

*Figure 5 continued*

indicated time. Cell lysates were subjected to Western blots using the indicated antibodies. (E) Cultured S2 cells were treated with 50 μg/ml CHX for 0 (lanes 1, 3 and 5) or 150 min (lanes 2, 4 and 6). Cells were infected with DCV immediately after CHX addition (lanes 3 and 4) or 6 hr before CHX addition (lanes 5 and 6). Cell lysates were then prepared and subjected to western blots using the indicated antibodies. (F–G) Cultured S2 cells were mock infected (F) or infected with DCV (MOI = 5) (G) for 6 hr and then treated with 50 μg/ml CHX for the indicated periods. Cell lysates were prepared and subjected to western blots using the indicated antibodies. For (A, C–G), the values listed below the blots indicate the relative NTAN1 or EGFP protein levels following α-Tubulin normalization using Quantity One software. The protein level shown in lanes 1 was defined as 100% (or 1). (H) The relative levels of NTAN1 protein shown in (F) and (G) were plotted. All data represent means and SD of three independent experiments.
DOI: https://doi.org/10.7554/eLife.30590.011

The following source data and figure supplement are available for figure 5:

**Source data 1.** Quantification data for *Figure 5*.
DOI: https://doi.org/10.7554/eLife.30590.013
**Figure supplement 1.** Viral infection promotes the accumulation of NTAN1 after 12 h.p.i.
DOI: https://doi.org/10.7554/eLife.30590.012

Interestingly, NTAN1$^{4KA}$ is significantly more stable than NTAN1$^{WT}$ (*Figures 6D, F and H*); additionally, unlike NTAN1$^{WT}$ (*Figure 6A*, lanes 1 vs. 5), blocking the proteasome by MG-132 treatment did not show any effect on the protein level of NTAN1$^{4KA}$ in the absence of viral infection (*Figure 6C*, lanes 1 vs. 4), suggesting that one or all of these lysine residues and/or polyubiquitylation have some contribution to the protein stability of NTAN1. Moreover, our results showed that NTAN1$^{WT}$ but not NTAN1$^{4KA}$ can be polyubiquitylated, while the polyubiquitylation of NTAN1 was not affected by viral infection (*Figure 6—figure supplement 2A*).

Next, we constructed four mutants of NTAN1, that is K40A, K63A, K134A and K186A, to determine which lysine residue(s) are most responsible for the polyubiquitylation-dependent degradation of NTAN1. Our data showed that, similarly with NTAN1$^{WT}$, the MG-132 treatment dramatically enhanced the protein level of exogenously expressed NTAN1$^{K40A}$, NTAN1$^{K63A}$ or NTAN1$^{K134A}$ in the absence of viral infection (*Figure 6—figure supplement 2B,C and D*, lanes 1 vs. 4). On the other hand, like NTAN1$^{4KA}$, NTAN1$^{K186A}$ is resistant to degradation in the absence of viral infection (*Figure 6C* and *Figure 6—figure supplement 2E*, lanes 1 vs. 4), indicating that K186 is most responsible for the ubiquitylation-dependent degradation of NTAN1 in the absence of viral infection.

Altogether, our data showed that NTAN1 can be polyubiquitylated, and degraded by both ubiquitylation-dependent and -independent degradation pathways. While both NTAN1 degradation pathways are dependent on proteasome, the virus-induced NTAN1 degradation is independent of ubiquitylation (as illustrated in *Figure 6I*).

## Virus-induced NTAN1 degradation inhibits apoptosis and benefits viral replication

We ought to investigate the role of virus-induced NTAN1 degradation on apoptosis and viral replication. Our previous data have shown that the loss of NTAN1 can prevent the degradation of cleaved DIAP1 (*Figure 4C*). Here, we ectopically expressed HA-tagged NTAN1 in virally infected cells. Our results showed that the ectopic expression of HA-NTAN1 partially restored the expression of NTAN1, resulting in the almost elimination of both full-length and caspase-cleaved forms of DIAP1 at 15 and 18 h.p.i. (*Figure 7A*). Consequently, in the context of viral infection, the partial restoration of NTAN1 expression significantly promoted apoptosis (*Figure 7B*) and the relative caspase activity in cells (*Figure 7C*). Furthermore, the partial restoration of NTAN1 expression also significantly restricted viral RNA replication at 18 h.p.i. (*Figure 7D*). Moreover, the knockdown of NTAN1 inhibited virus-induced apoptosis and enhanced DCV replication (*Figure 7—figure supplement 1A and B*). Altogether, these data indicate that virus-induced NTAN1 degradation can inhibit apoptosis and benefit viral replication.

## Discussion

In this study, we demonstrate that a picorna-like virus can induce apoptosis, and the virus-induced apoptosis plays an antiviral role in *Drosophila*. Strikingly, we uncovered that viral infection is able to induce the degradation of NTAN1, a key component of the N-end rule degradation pathway, via an

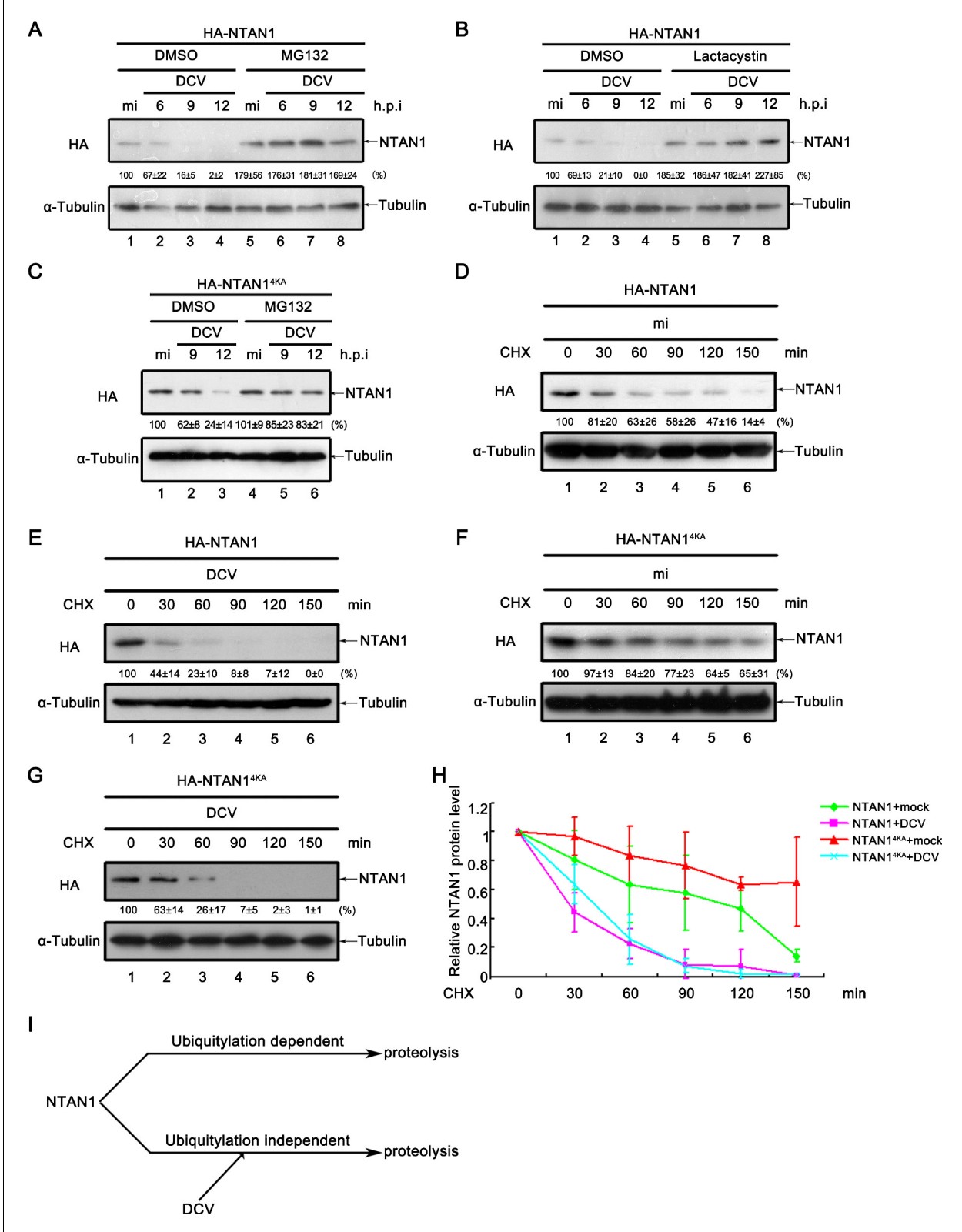

**Figure 6.** Viral infection promotes the degradation of NTAN1 via the proteasome pathway. (A–B) Cultured S2 cells were transfected with plasmid expressing HA-NTAN1, and then treated with DMSO, MG132 (A) or lactacystin (B) as indicated. After that, cells were mock infected for 12 hr or infected with DCV (MOI = 5) for indicated time. (C) Cultured S2 cells were transfected with plasmid expressing HA-NTAN1⁴ᴷᴬ, and then treated with DMSO or MG132 as indicated. After that, cells were mock infected for 12 hr or infected with DCV (MOI = 5) for indicated time. (D–G) Cultured S2 cells were

*Figure 6 continued on next page*

Figure 6 continued

transfected with plasmid expressing HA-NTAN1 (**D and E**) or HA-NTAN1[4KA] (**F and G**) and then mock infected (**D and F**) or infected with DCV (MOI = 5) (**E and G**) for 6 hr. After that, cells were treated with 50 µg/ml CHX for the indicated periods. (**A–G**) Cell lysates were subjected to western blots using the indicated antibodies. The values listed below the blots indicate the relative NTAN1 protein levels compared to lane 1 following α-tubulin normalization using Quantity One software. (**H**) The relative levels of NTAN1 protein shown in (**D**), (**E**), (**F**) and (**G**) were plotted. All data represent means and SD of three independent experiments. (**I**) Proposed model of NTAN1 degradation strategies.
DOI: https://doi.org/10.7554/eLife.30590.014

The following source data and figure supplements are available for figure 6:

**Source data 1.** Quantification data for *Figure 6.*
DOI: https://doi.org/10.7554/eLife.30590.017

**Figure supplement 1.** Amino acid sequence analyses of NTAN1 proteins in different organisms.
DOI: https://doi.org/10.7554/eLife.30590.015

**Figure supplement 2.** K186 is the critical lysine residue for ubiquitylation of NTAN1.
DOI: https://doi.org/10.7554/eLife.30590.016

ubiquitylation-independent proteasome pathway in cells. The virus-induced degradation of NTAN1 caused the accumulation of caspase-cleaved, shorter form of DIAP1 by inhibiting its N-terminal Asn deamidation, resulting in the suppression of apoptosis and the enhancement of viral replication (as illustrated in *Figure 8*).

The apoptotic pathway is recognized as an antiviral defense mechanism (*Everett and McFadden, 1999*), while various viruses employ their own ways to evade apoptosis. For example, SV40 large T antigen can bind to and inactivate p53, the internal sensor of the apoptotic pathway (*Lane and Crawford, 1979*; *Linzer and Levine, 1979*); adenovirus E1B-19K simulates the anti-apoptotic regulator Bcl-2 (*Chiou et al., 1994*) and regulates the activity of p53 (*Lomonosova et al., 2005*); baculovirus P35 and P49 can block the activity of caspase (*Lannan et al., 2007*; *Zoog et al., 2002*), and P35 can also bind to and stabilize a cellular IAP (*Byers et al., 20162015*). Our current study uncovered that the infection by a picorna-like virus can suppress the N-end rule pathway by inducing the degradation of its key component NTAN1. This process causes the accumulation of caspase-cleaved DIAP1, which results in apoptosis inhibition and represents a novel mechanism of viral evasion of apoptosis.

It is interesting that the virus-induced degradation of NTAN1 is dependent of proteasome but independent of ubiquitylation (*Figure 6*). Protein degradation via the ubiquitin proteasome system has been extensively studied. In contrast, the mechanism of the ubiquitin-independent proteolytic activity of proteasomes is poorly understood. It has been reported that the ubiquitin-independent proteolytic activity of proteasomes is involved in the degradation of oxidized proteins, chemically unfolded proteins, and specific natively disordered proteins (*Baugh et al., 2009*; *Grune et al., 2003*; *Hoyt and Coffino, 2004*). It is worth to mention that several important regulatory proteins can be degraded by this mechanism (*Hoyt and Coffino, 2004*), including p21/Cip1 (*Sheaff et al., 2000*), IκBα (*Krappmann et al., 1996*), c-Jun (*Jariel-Encontre et al., 1995*) and p53 (*Tsvetkov et al., 2009*; *Tsvetkov et al., 2010*). The N-end rule pathway is normally recognized as an ubiquitin proteasome proteolytic system and employs specific E3 ubiquitin ligases. As a key component of the N-end rule pathway, NTAN1 can be degraded in an ubiquitin-independent manner suggests a connection between these distinct proteasome proteolytic mechanisms, which extends the knowledge about ubiquitin-independent proteolytic activity of proteasome. The future study by us or others should uncover how viral infection induces the ubiquitin-independent NTAN1 degradation.

The N-end rule pathway plays an important role in various biological processes. According to the different substrates, the N-end rule pathway can be grouped into three types, the Arg/N-end rule pathway targets proteins with N-terminal Arg residue, the Ac/N-end rule pathway targets proteins with N-terminal acetylated residues and the Pro/N-end rule pathway targets proteins with N-terminal Pro residue or a Pro at position 2 (*Varshavsky, 2011*; *Park et al., 2015*; *Kim et al., 2014b*; *Chen et al., 2017*). Previous studies have shown the involvement of N-end rule pathway in a large number of important cellular processes. Besides its function in regulating DIAP1 in *Drosophila*, the Arg/N-end rule pathway can also regulate the C-terminal fragments of the Scc1 cohesin subunit that are produced by separase and thus regulates chromosome stability (*Rao et al., 2001*). Moreover, the N-end rule pathway regulates the mammalian G protein signaling through degrading RGS (regulator of G protein signaling) proteins (*Davydov and Varshavsky, 2000*; *Lee et al., 2005*;

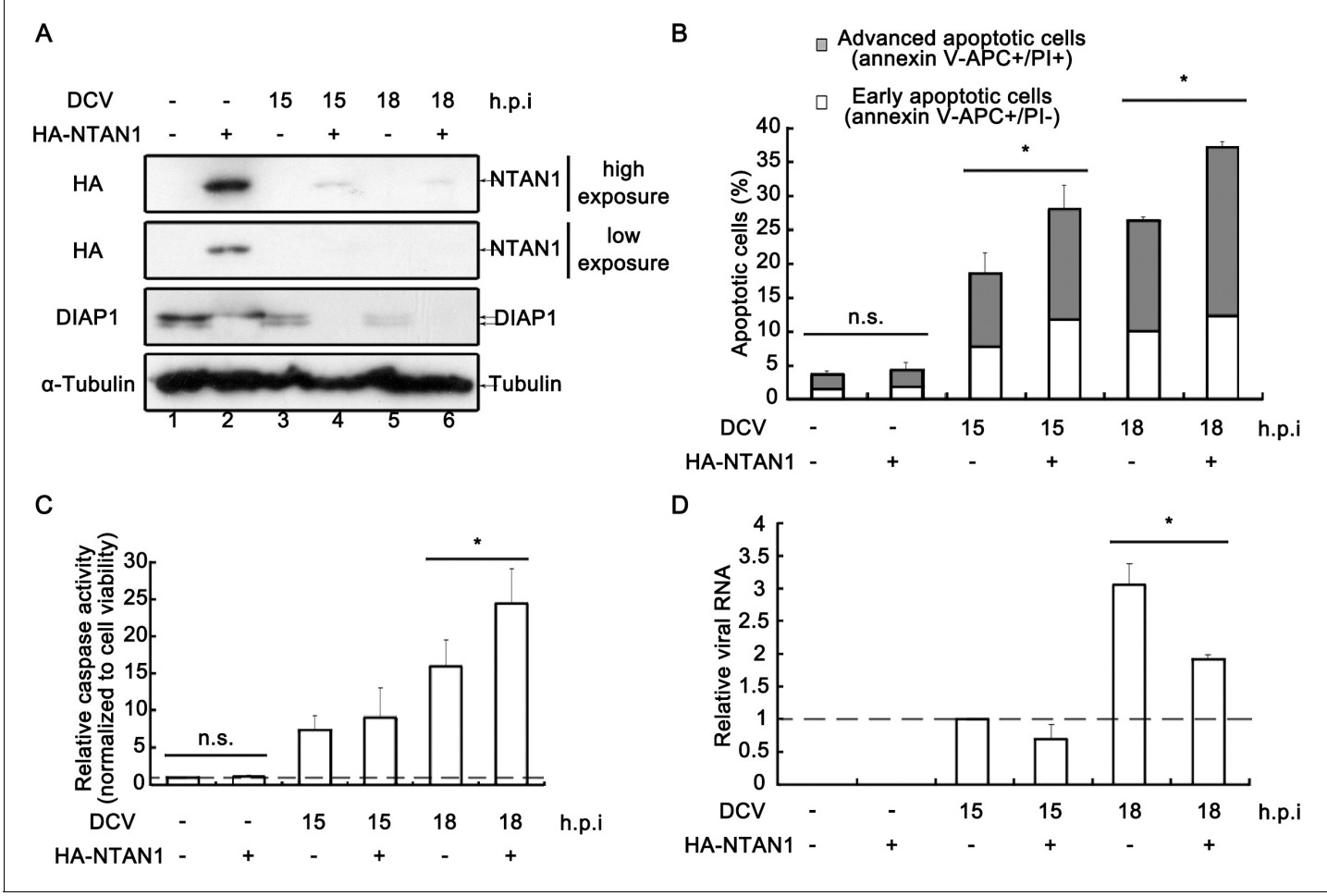

**Figure 7.** Restoring NTAN1 expression enhances virus-induced apoptosis and restricts viral replication in cells. (A–D) Cultured S2 cells were transfected with empty vector or plasmid expressing HA-NTAN1 as indicated, and then mock infected for 18 hr or infected with DCV (MOI = 5) for indicated time. (A) Cell lysates were prepared and subjected to western blots using the indicated antibodies. (B) The percentages of early apoptotic and late apoptotic cells were measured by Annexin-V-APC/PI double staining and flow cytometry assay ($n = 6$; *$p<0.05$ by two-tailed Student's $t$ test; error bars, s.d.). (C) The relative caspase activity was measured, and normalized to cell viability ($n = 3$; *$p<0.05$ by two-tailed Student's $t$ test; error bars, s.d.). (D) Total RNAs were extracted and then subjected to qRT-PCR assay of viral genomic RNA ($n = 3$; *$p<0.05$ by two-tailed Student's $t$ test; error bars, s.d.).

DOI: https://doi.org/10.7554/eLife.30590.018

The following source data and figure supplement are available for figure 7:

**Source data 1.** Quanification data for *Figure 7* and *Figure 7—figure supplement 1*.

DOI: https://doi.org/10.7554/eLife.30590.020

**Figure supplement 1.** Knockdown of NTAN1 inhibits virus-induced apoptosis and promotes viral replication.

DOI: https://doi.org/10.7554/eLife.30590.019

*Park et al., 2015*). In the N-end rule pathway, NTAN1 is the key component that regulates the half-life of a protein by identifying its N-terminal residue and initiating the process. It has been reported that NTAN1-deficient mice have neurological defects such as impairment of spontaneous activity and spatial memory (*Balogh et al., 2000*, *2001*). Our current study found that viral infection can induce NTAN1 degradation, resulting in the suppression of the N-end rule pathway and subsequent evasion of apoptosis.

It would be intriguing to find out how viral infection induces the degradation of NTAN1. Interestingly, we found that blocking protein synthesis by CHX before viral infection abolished the effect of DCV to induce NTAN1 degradation, indicating that viral protein synthesis and/or viral replication within infected cells, but not the input viral components, are responsible for this process. Viruses in

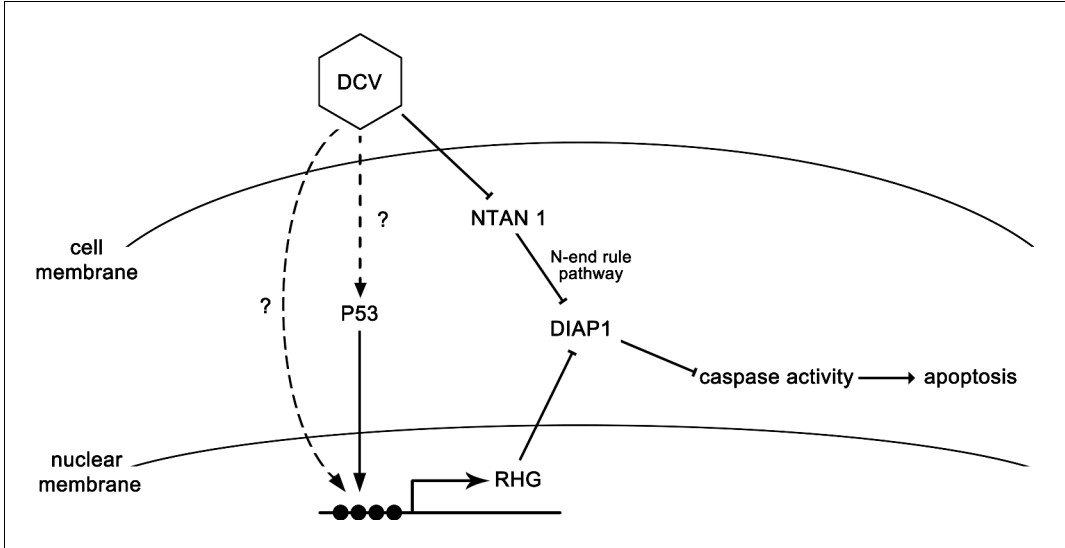

**Figure 8.** The model of apoptosis regulation during DCV infection. DCV infection induces RHG genes transcription and further induces apoptosis in *Drosophila*. On the other hand, DCV infection causes the degradation of NTAN1. This process suppresses the degradation of caspase-cleaved DIAP1 by inhibiting the N-end rule pathway, resulting in the suppression of apoptosis.

DOI: https://doi.org/10.7554/eLife.30590.021

the order *Picornavirales* encode a 3C or 3C-like (3 CL) protease that cleaves viral polyproteins. It has been reported that 3C proteases from multiple mammalian picornaviruses, such as foot-and-mouth disease virus (FMDV), Hepatitis A Virus, enterovirus 68, and enterovirus 71, are able to cleave and degrade host proteins to manipulate immune responses (*Wang et al., 2012*; *Yang et al., 2007*; *Xiang et al., 2014*; *Lei et al., 2013*), leading us to speculate whether DCV 3 CL protease could mediate NTAN1 degradation. However, we failed to observe any effect of exogenously expressed DCV 3 CL on the stability of NTAN1 (data not shown). It is also possible that other DCV proteins are responsible for the virus-induced NTAN1 degradation. Based on the sequence analyses, we have predicted the sequences and boundaries of DCV proteins. Unfortunately, the extreme difficulty to exogenously express other DCV proteins in *Drosophila* S2 cells prevented us from further examining these possibilities. In addition, viral infection can induce multiple intracellular signaling pathways, which may induce NTAN1 degradation.

In summary, our findings demonstrate for the first time that a virus can suppress the N-end rule pathway, and uncover a new mechanism for virus to evade apoptosis. Given the high conservation of the N-end rule pathway from prokaryotes to eukaryotes, it opens up the possibilities that this mechanism can also be employed by other viruses, particularly picornaviruses, to evade apoptosis and/or modulate other cellular processes, which are the targets of N-end rule pathway, in mammals or other organisms.

## Materials and methods

### Key resources table

| Reagent type (species) or resource | Designation | Source or reference | Identifiers | Additional information |
|---|---|---|---|---|
| cell line (*Drosophila melanogaster*) | S2 | ATCC | ATCC, Cat# CRL-1963; RRID: CVCL_Z232 | |
| antibody | anti-Flag M2 (mouse monoclonal) | Sigma | Sigma,Cat# F1804; RRID: AB_262044 | 1:2000 |
| antibody | anti-myc (mouse monoclonal) | MBL | MBL, Cat# M192-3; RRID: AB_11160947 | 1:2000 |

*Continued on next page*

*Continued*

| Reagent type (species) or resource | Designation | Source or reference | Identifiers | Additional information |
|---|---|---|---|---|
| antibody | anti-HA (mouse monoclonal) | ProteinTech | ProteinTech, Cat# 66006–1-Ig | 1:5000 |
| antibody | anti-α-Tubulin (mouse monoclonal) | ProteinTech | ProteinTech, Cat# 66031–1-Ig; RRID: AB_11042766 | 1:3000 |
| antibody | anti-DIAP1 (goat polyclonal) | Santa Cruz Biotechnology | Santa Cruz Biotechnology, Cat# sc-32414; RRID: AB_639332 | 1:200 |
| antibody | HRP-conjugated anti-GFP | ProteinTech | ProteinTech, Cat# HRP-66002 | 1:5000 |
| antibody | anti-ubiquitin (mouse monoclonal) | Cell Signaling Technology | Cell Signaling Technology, Cat# 3936; RRID:AB_331292 | 1:2000 |
| commercial assay or kit | Annexin-V-APC/PI double staining kit | BioLegend | BioLegend, Cat# 640932 | |
| commercial assay or kit | TUNEL staining kit | Roche | Roche, Cat# 11684817910 | |
| commercial assay or kit | CellTiter-Blue Cell Viability kit | Promega | Promega, Cat# G8080 | |
| commercial assay or kit | Caspase-Glo 3/7 kit | Promega | Promega, Cat# G8090 | |
| other | z-VAD-FMK | MedChem Express | MedChem Express, Cat# HY-16658 | 20 µM |
| other | CHX | Sigma | Sigma, Cat# C7698 | 50 µg/ml |
| other | MG-132 | Sigma | Sigma, Cat# C2211 | 50 µM |
| other | lactacystin | Merck | Merck, Cat# 426100 | 10 µM |

## Fly stocks and DCV oral infection

All flies used were 3- to 5-day-old adults reared at 25°C on a standard cornmeal/yeast diet. For each group, adult flies were randomly allocated and the sample size was chosen according to previous study (*Wang et al., 2015*). The p53 loss-of-function allele 5A-1–4 was obtained from the Blomington stock center. The $w^{1118}$ fly line used for control was obtained from Institute of Genetics and Developmental Biology, Chinese Academy of Sciences (Beijing, China).

DCV oral infections were performed on 3–6 days-old flies. Flies were randomly allocated into mock infection and DCV infection groups. For DCV oral infection, 2 ml of a mix of 25% virus extract ($10^{11.5}$ $TCID_{50}$/ml), 25% of yeast and 50% of standard cornmeal/yeast diet were loaded on a $1 \times 5$ cm filter paper. Each treated filter paper was placed in the bottom of an empty plastic vial. For the first 3 days, 30 flies per vial were placed and fed for 24 hr at 25°C, and then moved to a new vial containing filter paper treated as above. After that, we transferred the flies to new vials containing standard cornmeal/yeast diet. For mock oral infections flies, we used PBS instead of DCV extract to load the filter paper.

## Plasmid and in vitro transcription of RNA or dsRNA

The *Drosophila* inducible expression system vector, pAc5.1/V5-His B (Invitrogen, Carlsbad, CA), was used to construct plasmid that express protein in *Drosophila* S2 cells. The *diap1* or *ntan1* ORF was amplified from fly cDNAs, kindly provided by Dr. Jianquan Ni (Tsinghua University, Beijing, China), by polymerase chain reaction (PCR). The *diap1* ORF or its mutant carrying a myc tag at its 5'-end and a HA tag at its 3'-end was cloned into the *EcoR-Xho* site of the pAc5.1/V5-His B vector downstream of the *Drosophila* actin 5C promoter. The *ntan1* ORF or its mutant carrying a HA tag at its 5'-end was cloned into the *EcoR-Xho* site of the pAc5.1/V5-His B vector downstream of the *Drosophila* actin 5C promoter.

The dsRNAs used for RNAi were transcribed in vitro from the PCR products using T7 RNA polymerase (Promega) for 4 hr. The complete ORF of *drice* and *dronc*, nucleotides 1–400 of *egfp* ORF, nucleotides 1–422 of *ntan1* ORF and nucleotides 1–415 of *ate1* ORF were designed for generation of dsRNAs.

## Cell line

S2-ATCC cells (RRID: CVCL_Z232) was obtained from American Type Culture Collection (ATCC). Its identity was confirmed by visual inspection of the cell morphology and its growth kinetics in Schneider's insect medium (Sigma)/10% fetal bovine serum (FBS). A mycoplasma test is usually not done for S2 cells (*Berndt et al., 2017*).

The cell numbers were counted by using Luna automated cell counter (Logos Biosystems, Anyang-si, South Korea), according to the manufacturer's instruction.

## Transfection

The DNA or dsRNA transfection was performed as previously described (*Qiu et al., 2011*). In brief, *Drosophila* S2 cells were plated in six-well plates and grown overnight to reach 80% confluence (about $3 \times 10^6$ cells per well). After that, DNA plasmid or dsRNA was transfected into the cells using FuGene HD transfection reagent (Roche), according to the manufacturer's protocol. In addition, for transfecting same plasmid in multiple wells, to ensure the equal transfection, cells cultured in a 100 mm plate were firstly transfected. After 24–36 hr, the transfected cells were randomly divided into six or eight wells of six-well plate, and cultured for ~6 more hrs to reach 80% confluence (about $3 \times 10^6$ cells per well). The cells were then subjected to viral infection or other treatments according to experimental requirements.

## Inhibitors

The pancaspase inhibitor z-VAD-FMK (MedChemExpress, NJ, USA) was supplemented at 20 µM. The protein synthesis inhibitor CHX (Sigma) was supplemented at 50 µg/ml. The proteasome inhibitor MG-132 (Sigma) was used at 50 µM. The proteasome inhibitor lactacystin (Merck) was supplemented at 10 µM.

## Western blots, immunoprecipitation (IP) and antibodies

Cultured S2 cells were harvested, and then lysed in radio-immunoprecipitation assay (RIPA) buffer. The cell lysates were then subjected to 10% SDS-PAGE, followed by western blots according to our standard procedures (*Wang et al., 2013*). All western blots experiments have been independently repeated at least three times. The quantification of western blots was done via densitometry by using Bio-Rad Quantity One software. Total protein loads were determined by using Coomassie Brilliant Blue R250 staining (Thermo Fisher).

The anti-Flag M2 mouse monoclonal antibody (Sigma, F1804) and anti-myc mouse monoclonal antibody (MBL, M192-3) were used at a dilution of 1:2000. The anti-HA mouse monoclonal antibody (ProteinTech, 66006–1-Ig) was used at a dilution of 1:5000. The anti-α-tubulin mouse monoclonal antibody (ProteinTech, 66031–1-Ig) was used at a dilution of 1:3000. The anti-DIAP1 goat polyclonal antibody (Santa Cruz Biotechnology, sc-32414) was used at a dilution of 1:200. The HRP-conjugated anti-GFP antibody (ProteinTech, HRP-66002) was used at a dilution of 1:5000. The anti-ubiquitin mouse monoclonal antibody (Cell Signaling Technology, #3936) was used at a dilution of 1:2000. The anti-NTAN1 polyclonal antibody was raised in rabbits against peptide GGYRDAKGYGEDVF (GenScript antibody service, Nanjing, China) and used at a dilution of 1:2500. The anti-ATE1 polyclonal antibody was raised in rabbits against peptide LGDSASYSTKSLTQ (GenScript antibody service) and used at a dilution of 1:2500.

IP assays were conducted according to our standard protocol (*Qi et al., 2011*). Proteins were extracted from the precipitates and then subjected to 10% SDS-PAGE and western blots.

## Quantitative reverse transcription-PCR (qRT-PCR)

Total RNA was extracted from $3 \times 10^6$ cells by using TRIzol reagent (TaKaRa Bio) and treated by RQ1 RNase-free DNase I (Promega) to remove DNAs as previously described (*Wang et al., 2013*). qRT-PCR were performed using SuperReal PreMix Plus kit (TIANGEN), according to the manufacturer's protocol. Gene-specific primers used for PCR amplification or qRT-PCR were listed below.

Hid For CTAAAACGCTTGGCGAACTT; Hid Rev CCCAAAAATCGCATTGATCT; Reaper For ACGGGGAAAACCAATAGTCC; Reaper Rev TGGCTCTGTGTCCTTGACTG; Grim For CAATA TTTCCGTGCCGCTGG; Grim Rev CGTAGCAGAAGATCTGGGCC; DIAP1 For CCCCAGTA TCCCGAATACGC; DIAP1 Rev TCTGTTTCAGGTTCCTCGGC; ATE1 For GCATACTTCGCCGCA

TAAATCG; ATE1 Rev CTATGGCGTAATCGGCATCGG; NTAN1 For GTGCTCGTGCTGAATGGTG; NTAN1 Rev CGTAGTCTCTGTAGACGGGATG; DCV For TCATCGGTATGCACATTGCT; DCV Rev CGCATAACCATGCTCTTCTG; Rp49 For AAGAAGCGCACCAAGCACTTCATC; Rp49 Rev TCTGTTG TCGATACCCTTGGGCTT.

## Flow cytometry

Cell death was assessed by Annexin-V-APC/PI double staining (BioLegend) following manufacturer's instructions. After acquisition by flow cytometry (Beckman Coulter), data were analyzed and imaged with FCS Express 5 Plus (De Novo Software) with adapted settings.

## TUNEL assay

Detection of apoptotic cells using TUNEL staining (Roche) was performed following manufacturer's instructions. In the same experiment, detection of DNA using DAPI staining (Sigma) was performed following manufacturer's instructions.

## Caspase activity assay

Caspase activity was measured using Caspase-Glo 3/7 kit (Promega) following manufacturer's instructions. In the same experiment, cell viability was measured using CellTiter-Blue Cell Viability kit (Promega) following manufacturer's instructions.

## Acknowledgements

We wish to thank Dr. Jianquan Ni (Beijing, China) for fly cDNAs and Dr. Qingfa Wu (Hefei, China) for DCV. We also wish to thank Dr. Hong-Bing Shu (Wuhan, China) for helpful discussion. This work was supported by the National Natural Science Foundation of China (NSFC) - Excellent Young Scientist Fund (No. 31522004 to XZ), the NSFC grant (No. 31600126 to ZW), the Newton Advanced Fellowship from the UK Academy of Medical Sciences and NSFC (No. 31761130075 to XZ), the Strategic Priority Research Program of Chinese Academy of Sciences (No. XDPB0301 to XZ), the National Basic Research Program of China (No. 2014CB542603 to XZ), the National High-Tech R&D Program of China (No. 2015AA020939 to XZ), the Natural Science Foundation of Hubei for Distinguished Scientist (No. 2016CFA045 to XZ), and the National Science Foundation for Post-doctoral Scientists of China (No. 2016M592378 to ZW).

## Additional information

### Funding

| Funder | Grant reference number | Author |
|---|---|---|
| National Natural Science Foundation of China | No. 31522004 | Xi Zhou |
| National Natural Science Foundation of China | No. 31600126 | Zhaowei Wang |
| Academy of Medical Sciences | No. 31761130075 | Xi Zhou |
| National Basic Research Program of China | No. 2014CB542603 | Xi Zhou |
| Chinese Academy of Sciences | Strategic Priority Research Program. No. XDPB0301 | Xi Zhou |
| Natural Science Foundation of Hubei Province | No. 2016CFA045 | Xi Zhou |
| National High-Tech R&D Program of China | No. 2015AA020939 | Xi Zhou |

The funders had no role in study design, data collection and interpretation, or the decision to submit the work for publication.

## Author contributions
Zhaowei Wang, Conceptualization, Data curation, Software, Formal analysis, Funding acquisition, Validation, Investigation, Visualization, Methodology, Writing—original draft, Writing—review and editing; Xiaoling Xia, Conceptualization, Data curation, Formal analysis, Validation, Investigation, Visualization, Methodology, Writing—original draft, Writing—review and editing; Xueli Yang, Data curation, Validation, Investigation, Writing—review and editing; Xueyi Zhang, Yujie Liu, Investigation, Writing—review and editing; Yongxiang Liu, Di Wu, Yuan Fang, Formal analysis, Investigation, Writing—review and editing; Jiuyue Xu, Validation, Investigation, Writing—review and editing; Yang Qiu, Formal analysis, Investigation, Methodology, Writing—review and editing; Xi Zhou, Conceptualization, Data curation, Formal analysis, Supervision, Funding acquisition, Investigation, Writing—original draft, Project administration, Writing—review and editing

## Author ORCIDs
Xi Zhou (iD) https://orcid.org/0000-0002-3846-5079

## Decision letter and Author response
Decision letter https://doi.org/10.7554/eLife.30590.024
Author response https://doi.org/10.7554/eLife.30590.025

## Additional files

### Supplementary files
• Transparent reporting form
DOI: https://doi.org/10.7554/eLife.30590.022

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
