## [Decision Letter]

Thank you for submitting your article "A Picorna-like Virus Suppresses the N-end Rule Pathway to Inhibit Apoptosis" for consideration by *eLife*. Your article has been favorably evaluated by Wenhui Li (Senior Editor) and three reviewers, one of whom, Yong Tae Kwon (Reviewer #1), is a member of our Board of Reviewing Editors.

The reviewers have discussed the reviews with one another and the Reviewing Editor has drafted this decision to help you prepare a revised submission.

The N-end rule pathway has not been implicated in viral replication in a major way. A previous study by Ditzel et al. (03' NCB) showed that DIAP1, a *Drosophila* inhibitor of apoptosis, is cleaved by caspase, exposing N-terminal Asn21 which is subsequently deamidated by NTAN1 and arginylated by ATE1 of the N-end rule pathway, leading to ubiquitination by UBR box N-recognins and proteasomal degradation. In this study, the authors show that viral infection induces the degradation of NTAN1 leading to the metabolic stabilization of Asn21-DIAP1 (a short form) and, thus, inhibiting apoptosis of host cells. Moreover, the authors suggest that virus-induced degradation of NTAN1 is mediated by the proteasome in a manner independent from ubiquitination. The topics are of significant importance and novelty in the field of proteolysis and viral infection. Unfortunately, this study suffers from several major issues as summarized below. Overall, this manuscript needs major revision with several additional experiments.

Major Comments:

1) To determine the metabolic stabilities of DIAP1 and NTAN1 proteins, the authors measured their steady state levels using only immunoblotting analysis. The results from immunoblotting analysis are not sufficient to draw conclusions on protein degradation and should be validated using in vivo degradation assays such as pulse chase analysis. If pulse chase assay is technically impossible, an alternative is cycloheximide degradation assays.

2) The authors claim that virus induced degradation of NTAN1 is mediated by the proteasome without ubiquitination. In general, such negative conclusion with ubiquitination requires more rigorous validation. For example, in the second paragraph of the subsection “Viral infection promotes the degradation of NTAN1 via the proteasome pathway” (Figure 6), the authors claim that the mutation of all four lysine residues does not inhibit the degradation of NTAN and suggested that the virus-induced NTAN1 degradation is independent of ubiquitination. This immunoblotting analysis is of poor quality. The decay of NTAN1-4KA should be compared with wild-type NTAN1. In addition, the results should be validated using pulse chase analysis or at least cycloheximide degradation assay. To this reviewer, NTAN1-4KA seems to be significantly stabilized at 9 h.p.i. when compared with all four single lysine mutants (Figure 6—figure supplement 2).

3) The authors claim that viral infection induces the degradation of NTAN1 but did not examine the decay rate of endogenous NTAN1 in normally growing cells. It is necessary to compare the degradation rates of NTAN1 under two conditions with or without viral infection using in vivo degradation assays.

4) One of the major weaknesses of this study is the lack of any mechanistic insight into which viral components are involved in targeting NTAN1. It seems to be reasonably straightforward to consider the effects of the proteins introduced by the virus and perform at least basic tests to see how they could potentially affect NTAN1 stability. The authors offer no insights into this, not even a discussion of the possibilities. While I understand that detailed experiments into this direction may be difficult, I believe the paper would be considerably strengthened by at least a discussion or a model.

5) Western blots in all panels should be quantified, with sufficient number of repeats. For many panels, only one repeat is shown, making it unclear even how many experiments were done. It is essential to quantify data in triplicates, given the variability of the method itself.

6) In many panels, a consistent decrease of tubulin level with time post infection is seen. It appears that the virus has an effect on microtubules, which is probably not surprising but this raises the issues of loading controls. A potential loading control could be total protein load, which could also be illustrated by Coomassie Blue staining. There are also other commonly used loading controls like GAPDH or actin – which are, of course, also subject to change per virus infection. This should be addressed.

7) Along the same lines, quantification of the levels of transfected protein should be accompanied by qPCR data showing the transfection levels, since these can considerably affect the result between cultures. A "quick and dirty" way around it is to split the culture equally after transfection into the wells/dishes used for the collection of the time points. I did not find any experimental details about this in the manuscript.

8) In Figure 7, additional NTAN1 enhances virus-induced apoptosis and reduced viral replication in cell. How about in NTAN1 knockdown cells? In Figure 4, ATE1 and NTAN1 knockdown cells by dsRNA shows accumulation of cleaved form of DAIP1. It is important to examine the caspase level or apoptotic state and DCV gRNA in NTAN1 knockdown cells upon DCV infection.

9) Mechanism of antiviral function by apoptosis is not clear. Increased level of DCV gRNA in cells could be interpreted by many reasons. The viral RNA replication was enhanced in cells by anti-apoptotic DIAPI, viral budding pathway was blocked so that viral RNA was accumulated in the cells, or viral entry was more efficient than other groups, etc. Replicated RNA genome is packaged and newly packaged viral particles should be budded out into culture fluids. One feasible experiment is to analyze viral particles in cultured fluids by either RT-qPCR or any available methods in order to address the increased level of DCV gRNA means more infectious viable DCV.

---

## [Author Response]

Major Comments:1) To determine the metabolic stabilities of DIAP1 and NTAN1 proteins, the authors measured their steady state levels using only immunoblotting analysis. The results from immunoblotting analysis are not sufficient to draw conclusions on protein degradation and should be validated using in vivo degradation assays such as pulse chase analysis. If pulse chase assay is technically impossible, an alternative is cycloheximide degradation assays.

We sincerely appreciate and agree with this valuable comment. Following this suggestion, we conducted the CHX degradation assays for both DIAP1, NTAN1, and NTAN1-4KA (the mutant with all of the 4 lysine residues being mutated to Alanine). The data from the CHX degradation assays show that viral infection promoted the degradation of these proteins (new Figure 3—figure supplement 1; new Figure 5; new Figure 6; subsection “Viral infection promotes the accumulation of cleaved DIAP1 in cells”, first paragraph; subsection “Viral infection promotes the depletion of NTAN1 in the early stage of infection”, second paragraph and subsection “Viral infection promotes the degradation of NTAN1 via the proteasome pathway”, second paragraph).

2) The authors claim that virus induced degradation of NTAN1 is mediated by the proteasome without ubiquitination. In general, such negative conclusion with ubiquitination requires more rigorous validation. For example, in the second paragraph of the subsection “Viral infection promotes the degradation of NTAN1 via the proteasome pathway” (Figure 6), the authors claim that the mutation of all four lysine residues does not inhibit the degradation of NTAN and suggested that the virus-induced NTAN1 degradation is independent of ubiquitination. This immunoblotting analysis is of poor quality. The decay of NTAN1-4KA should be compared with wild-type NTAN1. In addition, the results should be validated using pulse chase analysis or at least cycloheximide degradation assay. To this reviewer, NTAN1-4KA seems to be significantly stabilized at 9 h.p.i. when compared with all four single lysine mutants (Figure 6—figure supplement 2).

We sincerely thank you for the valuable comments. Following this suggestion, we conducted the immunoblots in Figure 6 again with several repeats, and replaced it using a figure of better quality in the revised version (new Figure 6).

As we responded to the major comment #1, we have compared the decay of NTAN1-4KA with NTAN1 WT using CHX degradation assay. Our data show that viral infection could promote the degradation of both 4KA mutant and WT NTAN1 (new Figure 6; subsection “Viral infection promotes the degradation of NTAN1 via the proteasome pathway”, second paragraph), although NTAN1-4KA is more stable than WT NTAN1 in the absence of DCV infection (new Figure 6). In addition, based on our data and quantification in Figure 6, viral infection induced the degradation of NTAN1-4KA at 9 h.p.i, similarly with WT and other mutants. In fact, there are two ways to degrade NTAN1 (Ub-dependent and –independent) at the same time. NTAN1-4KA mutation can completely block the way of Ub-dependent degradation, while a single lysine mutation may not completely block this way. Thus, NTAN1-4KA is expectedly more stable than other single lysine mutants in the presence or absence of viral infection.

3) The authors claim that viral infection induces the degradation of NTAN1 but did not examine the decay rate of endogenous NTAN1 in normally growing cells. It is necessary to compare the degradation rates of NTAN1 under two conditions with or without viral infection using in vivo degradation assays.

We agree with this comment. Following this suggestion, we have examined the degradation rates of endogenous NTAN1 in normally growing cells in both the presence and absence of viral infection using the CHX degradation assay. And our data show that viral infection could induce endogenous NTAN1 degradation (new Figure 5; subsection “Viral infection promotes the depletion of NTAN1 in the early stage of infection”, second paragraph).

4) One of the major weaknesses of this study is the lack of any mechanistic insight into which viral components are involved in targeting NTAN1. It seems to be reasonably straightforward to consider the effects of the proteins introduced by the virus and perform at least basic tests to see how they could potentially affect NTAN1 stability. The authors offer no insights into this, not even a discussion of the possibilities. While I understand that detailed experiments into this direction may be difficult, I believe the paper would be considerably strengthened by at least a discussion or a model.

We sincerely appreciate with this valuable comment. Indeed, we have tried our best to figure out the exact mechanism by which DCV infection promotes NTAN1 degradation. To this end, we have tried to express individual DCV proteins in cultured cells to examine how they can potentially affect NTAN1 stability, as the reviewer also suggested. However, due to the limitation of current knowledge, the exact boundaries of these individual viral proteins in the DCV polyproteins are unclear (DCV genome contains two ORFs that encode two polyproteins, which are subsequently cleaved by DCV 3C-like protease into separate proteins). Based on the sequence analyses and the alignment with other picornaviruses and picorna-like viruses, we have predicted the sequences of these DCV proteins (1A, Helicase, RdRP, 3C-like, VP1, VP2, VP3, and VP4) and cloned them into expression vectors. Unfortunately, most of these viral proteins were extremely hard to be expressed in S2 cells (this phenomenon is not unusual for expressing many other viral proteins). We only successfully expressed DCV 3C-like protease (3CL) in S2 cells, which is an ideal candidate as picornavirus 3C protease has been reported to cleave many host proteins. However, we didn’t observe any effect of 3CL on NTAN1 stability (data not shown in the manuscript). Following this reviewer’s suggestion, we have discussed the possible mechanisms that may induce NTAN1 degradation in the revised manuscript (Discussion, fifth paragraph).

5) Western blots in all panels should be quantified, with sufficient number of repeats. For many panels, only one repeat is shown, making it unclear even how many experiments were done. It is essential to quantify data in triplicates, given the variability of the method itself.

We sincerely thank you for the comment. All of the Western blotting analyses had been independently repeated for at least three times. Following this suggestion, we have quantified the Western blotting data via densitometry in the revised figures and also described the method in the revised “Materials and methods” section (subsection “Western blots, immunoprecipitation (IP) and antibodies”, first paragraph).

6) In many panels, a consistent decrease of tubulin level with time post infection is seen. It appears that the virus has an effect on microtubules, which is probably not surprising but this raises the issues of loading controls. A potential loading control could be total protein load, which could also be illustrated by Coomassie Blue staining. There are also other commonly used loading controls like GAPDH or actin – which are, of course, also subject to change per virus infection. This should be addressed.

We sincerely appreciate and agree with this valuable comment. We consistently observed that viral infection could cause decrease of tubulin levels after 24-hr infection. Following this comment, we used total protein loads as loading controls by using Coomassie Blue staining (new Figure 3), and revised the “Materials and methods” section accordingly (subsection “Western blots, immunoprecipitation (IP) and antibodies”, first paragraph). In addition, from our own observation, GAPDH or actin may not be used as good loading control, as their commercial antibodies usually recognize many noise bands in the Western blots of *Drosophila* S2 cell extracts.

7) Along the same lines, quantification of the levels of transfected protein should be accompanied by qPCR data showing the transfection levels, since these can considerably affect the result between cultures. A "quick and dirty" way around it is to split the culture equally after transfection into the wells/dishes used for the collection of the time points. I did not find any experimental details about this in the manuscript.

We agree with this comment. Actually, we had already used the "quick and dirty" way suggested by the reviewers in our experiments. In brief, we firstly transfected cells in a 100-mm culture dish. After 24-36 hr transfection, the transfected cells were divided into six or eight wells of 6-well plate, and cultured for 6 more hours. The cells were then subjected to viral infection or other treatments according to experimental requirements. Based on this procedure, the transfection level or efficiency in different wells should be equal. The experimental procedures have been described in detail in the revised manuscript (subsection “Transfection”).

In addition, following this reviewers’ suggestion, we have quantified the transfection levels using qPCR, and our data show that using the above method, the transfection levels in different samples were similar (new Figure 3—figure supplement 2, subsection “Viral infection promotes the accumulation of cleaved DIAP1 in cells”, third paragraph).

8) In Figure 7, additional NTAN1 enhances virus-induced apoptosis and reduced viral replication in cell. How about in NTAN1 knockdown cells? In Figure 4, ATE1 and NTAN1 knockdown cells by dsRNA shows accumulation of cleaved form of DAIP1. It is important to examine the caspase level or apoptotic state and DCV gRNA in NTAN1 knockdown cells upon DCV infection.

We sincerely thank the reviewers for the comment. Following the suggestion, we have examined the effect of NTAN1 knockdown, and our data show that its knockdown inhibited virus-induced apoptosis and also enhanced DCV replication (new Figure 7—figure supplement 1, subsection “Virus-induced NTAN1 degradation inhibits apoptosis and benefits viral replication”).

9) Mechanism of antiviral function by apoptosis is not clear. Increased level of DCV gRNA in cells could be interpreted by many reasons. The viral RNA replication was enhanced in cells by anti-apoptotic DIAPI, viral budding pathway was blocked so that viral RNA was accumulated in the cells, or viral entry was more efficient than other groups, etc. Replicated RNA genome is packaged and newly packaged viral particles should be budded out into culture fluids. One feasible experiment is to analyze viral particles in cultured fluids by either RT-qPCR or any available methods in order to address the increased level of DCV gRNA means more infectious viable DCV.

We are grateful for this comment. Following this suggestion, we have determined the levels of viral particles in cultured fluids by qPCR. Our data show that overexpressing DIAP1 or knocking down caspases did not result in the decrease of the level of viruses in cultured fluids (new Figure 2, subsection “Inhibition of apoptosis enhances viral replication in cells and adult flies”, first paragraph), confirming that viral RNA replication was enhanced in apoptosis-inhibited cells.